# DNA end resection requires constitutive sumoylation of CtIP by CBX4

Isabel Soria-Bretones[1,2], Cristina Cepeda-García[2], Cintia Checa-Rodriguez[1,2], Vincent Heyer[3,4,5,6], Bernardo Reina-San-Martin[3,4,5,6], Evi Soutoglou[3,4,5,6] & Pablo Huertas [1,2]

DNA breaks are complex DNA lesions that can be repaired by two alternative mechanisms: non-homologous end-joining and homologous recombination. The decision between them depends on the activation of the DNA resection machinery, which blocks non-homologous end-joining and stimulates recombination. On the other hand, post-translational modifications play a critical role in DNA repair. We have found that the SUMO E3 ligase CBX4 controls resection through the key factor CtIP. Indeed, CBX4 depletion impairs CtIP constitutive sumoylation and DNA end processing. Importantly, mutating lysine 896 in CtIP recapitulates the CBX4-depletion phenotype, blocks homologous recombination and increases genomic instability. Artificial fusion of CtIP and SUMO suppresses the effects of both the non-sumoylatable CtIP mutant and CBX4 depletion. Mechanistically, CtIP sumoylation is essential for its recruitment to damaged DNA. In summary, sumoylation of CtIP at lysine 896 defines a subpopulation of the protein that is involved in DNA resection and recombination.

[1] Departamento de Genética, Universidad de Sevilla, Sevilla 41080, Spain. [2] Centro Andaluz de Biología Molecular y Medicina Regenerativa-CABIMER, Universidad de Sevilla-CSIC-Universidad Pablo de Olavide, Sevilla 41092, Spain. [3] Institut de Génétique et de Biologie Moléculaire et Cellulaire, Illkirch 67404, France. [4] Institut National de la Santé et de la Recherche Médicale U964, Illkirch 67404, France. [5] Centre National de Recherche Scientifique UMR7104, Illkirch 67404, France. [6] Université de Strasbourg, Illkirch 67081, France. Correspondence and requests for materials should be addressed to P.H. (email: pablo.huertas@cabimer.es)

Maintenance of genomic integrity is an essential priority for all living organisms[1,2]. In humans, genomic instability predisposes the appearance of several rare diseases and cancer, and could have grave consequences for progeny[2]. A major cause of such instability is the erroneous repair of chromosome breaks[3]. To avoid this, the choice between different DNA double-strand break (DSB) repair pathways is tightly regulated, mainly through the DNA end processing known as DNA resection[4]. Indeed, DNA DSBs can be simply re-joined with little or no processing by the non-homologous end-joining (NHEJ) machinery[5]. However, when specific criteria are met, DNA resection is activated. Such DNA processing consists on the degradation of one strand of the DNA, with a 5′–3′ polarity, that leaves a tail of protruding ssDNA, which is immediately coated by the protecting complex RPA[4]. This resected DNA can then pair with a homologous sequence, prompting homologous recombination and blocking NHEJ[4,6]. Thus, resection acts as a molecular switch that regulates how a DNA break will be repaired[4]. Hence, DNA end resection is a tightly regulated mechanism. The actual network of signals controlling resection is still poorly understood. A critical factor is the protein known as CtIP, a multifunctional protein that receive and somehow decipher cellular and environmental cues to activate resection[7]. For DNA end resection, especially relevant is the carboxy-terminal part of the protein, that is loosely related with its functional counterparts in budding and fission yeast, Sae2, and Cpt1, respectively[7]. Moreover, such region mediates the interaction with pro and anti-resection factors such as the MRN complex and CCAR2[8,9]. Finally, it is critical to interpret the cellular status and activate resection via post-translational modifications (PTMs) such as CDK phosphorylations[7].

The dynamic modulation of PTMs has been extensively reported to coordinate DNA repair and the global DNA damage response (DDR) (for reviews, see refs. [10–12]). Among them, in recent years sumoylation has emerged as a key element of the DDR. Sumoylation is a reversible process involving several enzymatic steps in which specific protein targets are modified by the covalent conjugation of a small peptide known as SUMO[13]. In mammals, the E1 SUMO-activating complex (SAE1 and SAE2) initiates the process, and a unique E2 conjugating enzyme (UBC9) binds the activated SUMO molecule to transfer it to the substrate together with one of the many E3 ligases present in the cell, which provide substrate specificity[13]. While numerous proteins have been proposed to act as a SUMO E3 ligase, the best characterized ones are the PIAS family[14], RanBP2[15], TOPORS, hMMS21[16], and CBX4 (also known as Pc2)[17]. PIAS1 and PIAS4 are important in the DDR and DSB repair pathways[18,19]. Despite the importance for the DDR, the specific roles of sumoylation in DNA end resection are still object of investigation. In budding yeast, proteomic studies have identified key resection factors as potential damage-dependent sumoylation targets[20,21]. Moreover, such global upregulation of sumoylation depends on the resection machinery itself[20,21]. In mammals, the effect of sumoylation in resection is less well understood. Recruitment of both pro- and anti-resection proteins requires sumoylation[18].

Here we show that the sumoylation of a single residue in the carboxy-terminal tail of the critical resection factor CtIP (Lys-896) by the SUMO E3 ligase CBX4 is essential for DNA resection, homologous recombination, and the maintenance of genomic stability. CBX4 depletion impaires CtIP constitutive sumoylation and DNA end processing. Importantly, mutating the single lysine at position 896 in CtIP recapitulates the CBX4-depletion phenotype, renders cells mildly sensitive to PARP inhibitor, reduces homologous recombination and increases genomic instability. Artificial fusion of CtIP with SUMO suppresses the effects of both the non-sumoylatable CtIP mutant and CBX4 depletion. Notably, CtIP sumoylation and its CDK-dependent phosphorylation at Threonine 847 are functionally connected. Mechanistically, sumoylation of CtIP is required for its recruitment to damaged DNA. Thus, CBX4-mediated constitutive sumoylation establishes the pool of CtIP that can be recruited to damaged chromatin, hence available for DNA resection and homologous recombination, effectively controlling DNA DSB repair.

## Results

**The SUMO ligase CBX4 is required for DNA end resection.** Given the relevance of sumoylation in the DDR, we wondered if such PTM might be directly involved in DNA repair and, more specifically, in DNA resection. To find SUMO E3 ligase(s) involved in DNA processing, we selected those that had either been previously linked to DDR or that are located in the nucleus. We knocked down each selected E3 ligase or, as a control, the key resection factor CtIP[8] in the human U2OS cell line, and assessed the proficiency of resection upon induction of DNA damage with ionizing radiation using RPA foci formation as a readout. Several of the single E3 ligase knockdowns reduced resection proficiency,

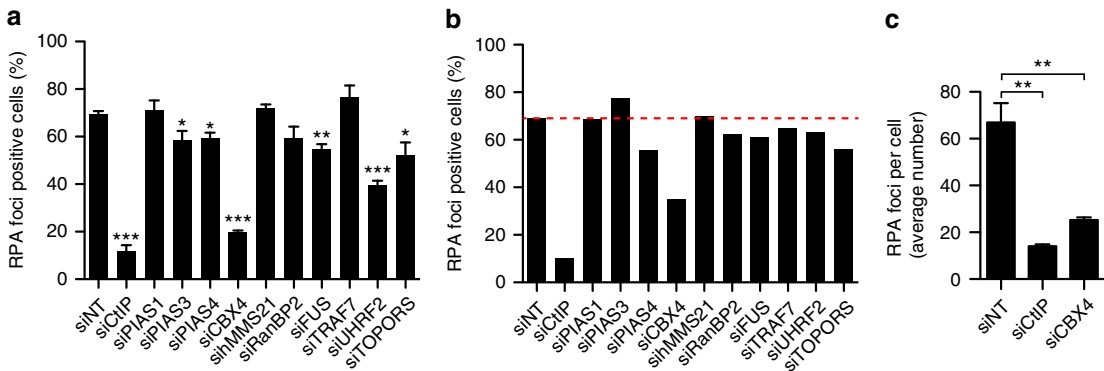

**Fig. 1** CBX4 is involved in DNA resection. **a** DNA resection proficiency measured as the percentage of RPA foci-positive cells upon exposing them to ionizing radiation (10 Gy). The average and SEM of three independent experiments are shown. Significance was determined by Student's *t*-test comparing each condition to siNT cells. *P < 0.05; **P < 0.01; ***P < 0.001. **b** Data represented in **a**, but normalized to cell cycle distribution (Supplementary Fig. 1d). Only S/G2 cells were used for normalization, with the S/G2 population of siNT cells considered as 1. **c** Average number of RPA foci per cell. Cells transfected with siRNAs against CtIP, CBX4 or a control siRNA (siNT) were treated as in **a** and the average of RPA foci per cell was scored using Metamorph software

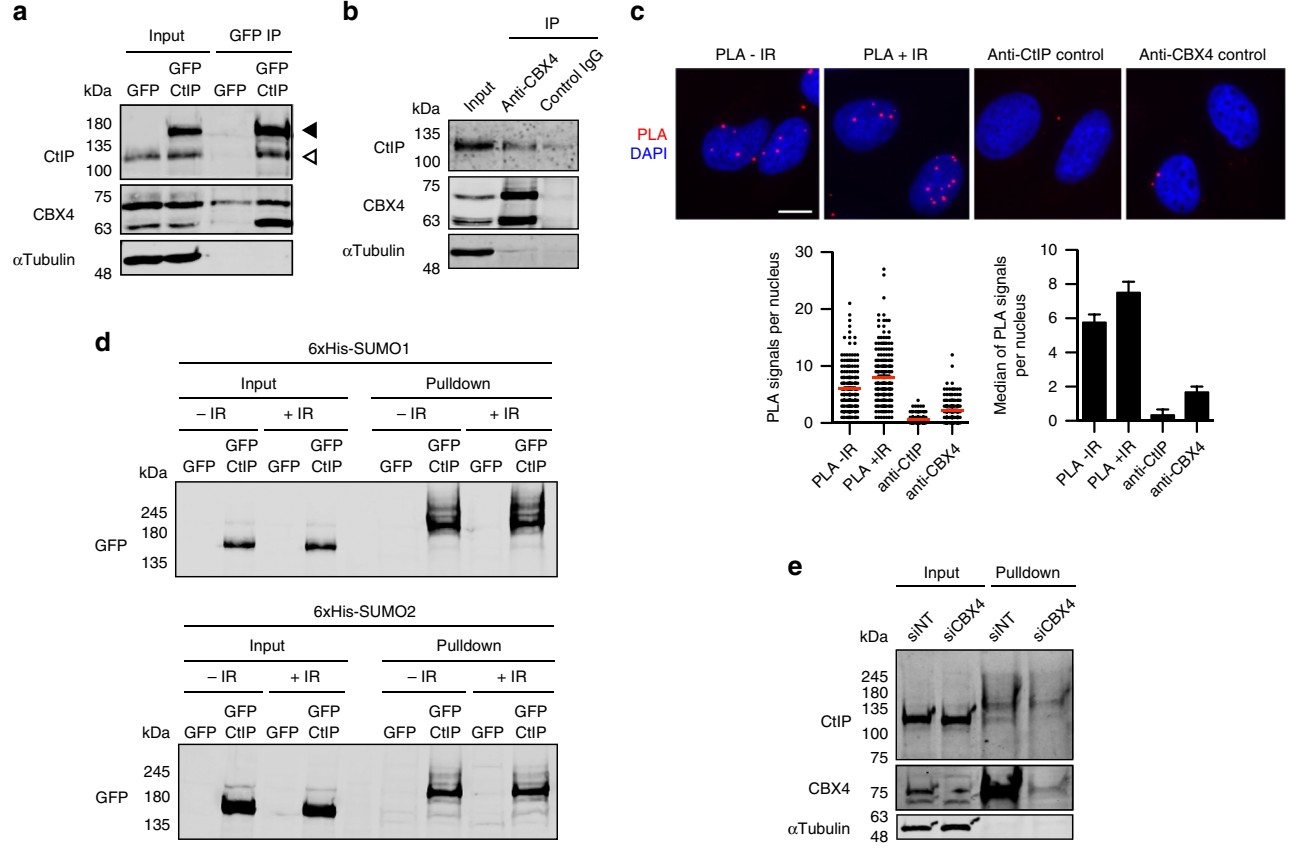

**Fig. 2** CBX4 interacts and sumoylates CtIP. **a** GFP immunoprecipitation (IP) of protein extracts derived from U2OS cells expressing GFP-CtIP or GFP were blotted with the indicated antibodies. Input represents 5% of the total amount of protein in the IP. Endogenous CtIP and GFP-CtIP are denoted with an empty arrow and a filled arrow, respectively. A representative experiment is shown. **b** Endogenous CBX4 was immunoprecipitated from U2OS cells using an antibody against CBX4. Protein samples were resolved in SDS-PAGE and blotted with the indicated antibodies. An IgG control was included to show binding specificity. Input represents 5% of the total amount of protein. **c** PLA assay between CtIP and CBX4 in undamaged cells (−IR) and cells 1 h after exposure to 10 Gy ionizing radiation (+IR). To assess specificity, cells were subjected to the same assay but with addition of only the CtIP or the CBX4 antibody. The *white scale bar* represents 10 μM. A representative experiment is plotted in the left graph; the median number of PLA signals is in *red*. The average and SEM of the median PLA signals corresponding to three independent experiments are plotted in the right graph. **d** HEK293T cells harboring GFP or GFP-CtIP and 6 × His-SUMO1 or -SUMO2 constructs, either untreated (−IR) or irradiated with 10 Gy (+IR), were used for His pulldowns, and protein samples were blotted with GFP antibody. Inputs represent 1% of the total amount of protein in the pulldowns. A representative experiment is shown. **e** U2OS cells stably expressing 10 × His-SUMO1 were transfected with siCBX4 or a control siRNA (siNT) and used for His pulldowns. Input represents 5% of the total amount of protein in the pulldown. A representative experiment is shown

measured as percentage of RPA positive cells (Fig. 1a, Supplementary Fig. 1a–c). However, extensive DNA resection is known to be limited to the S and G2 phases of the cell cycle[4, 22], and there were significant alterations of the cell cycle progression upon depletion of some E3s (Supplementary Fig. 1d). Thus, to minimize any effects of cell cycle arrest on RPA foci, we normalized the percentage of resection-proficient cells to the population of S/G2 cells in each case (Fig. 1b). Again, several single E3 knockdowns were able to slightly reduce resection proficiency, but only that of CBX4 caused a clear decrease in RPA foci formation regardless of cell cycle distribution, suggesting that CBX4 may play a specific, relevant role in DNA end resection. It is noteworthy that such normalization did not prompt the appearance of a resection phenotype upon downregulation of any of the E3, but actually reduced the effect in most of them (including CBX4). Even though, to be absolutely sure of the effect of CBX4 depletion on DNA end resection, we reanalysed RPA accumulation by a different parameter. We decided to perform a computer based quantification of the average number of RPA foci per cell upon CBX4 downregulation. Indeed, the average number of foci per cell was

also reduced in CBX4 depleted cells, validating our earlier observation (Fig. 1c).

**CBX4 constitutively sumoylates CtIP.** CBX4 is known to be recruited rapidly to DNA damage, and its depletion sensitizes cells to ionizing radiation[23]. CBX4 activity as a SUMO E3 ligase was firstly identified with CtBP1 and CtBP2 as targets[17, 24]. Strikingly, CtIP was also first identified as an interactor of CtBP[25]. Hence, we decided to test if CBX4 and CtIP interact. Indeed, we observed a DNA damage-independent interaction of both proteins by both co-immunoprecipitation and the proximity ligation assay (PLA), using both endogenous and tagged proteins (Fig. 2a–c). Such interaction occurred in the absence of an exogenous source of DNA damage.

This CBX4–CtIP interaction suggested that CtIP might be targeted for CBX4-mediated sumoylation in a constitutive, DNA damage-independent manner. We transiently co-expressed 6 × His-tagged-SUMO1 or -SUMO2 and GFP-CtIP in HEK293T cells and performed pulldowns under stringent conditions (with 8 M urea buffer). Indeed, several bands

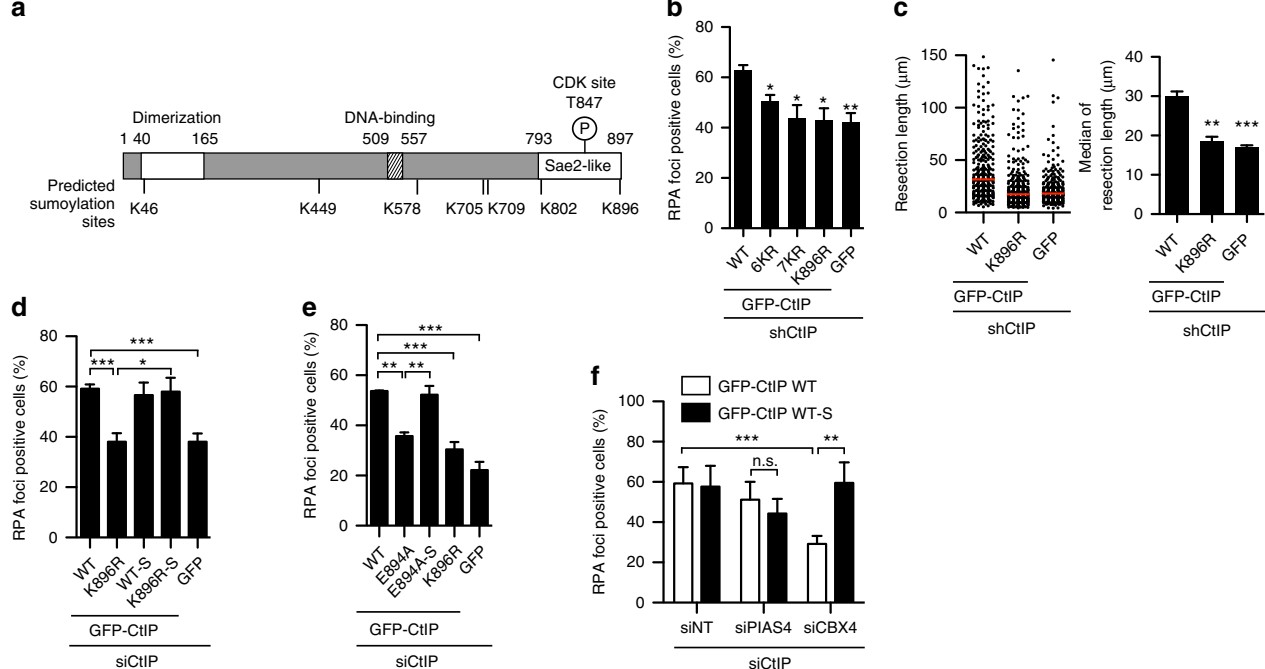

**Fig. 3** Sumoylation of K896 in CtIP is essential for DNA resection. **a** Diagram of human CtIP protein. The predicted sumoylation sites and the CDK-mediated phosphorylation site used in this study are depicted. **b** U2OS cells stably expressing the indicated variants of GFP-CtIP, or GFP as a control, and depleted of endogenous CtIP were irradiated with 10 Gy. RPA foci were detected 1 h after ionizing radiation. **c** Resection length measured with SMART assay using DNA fibers extracted from U2OS downregulated for endogenous CtIP and expressing GFP-CtIP, GFP-CtIP-K896R or GFP as control. The length of at least 200 fibers and the median length (in *red*) of a representative experiment in each case are plotted in the *left graph*. The average and SEM of the median length in three independent experiments are plotted in the *right graph*. In **b** and **c**, Student's *t*-test was performed in each condition compared to control. **d** Same as **b** but with cells expressing wild-type CtIP or the K896R mutant transcriptionally fused (WT-S and K896R-S) or not (WT, K896R) to SUMO1. **e** RPA foci formation was analyzed as in **b** and **d**, but in cells expressing the mutant E894A and its SUMO1 fusion (E894A-S). For **c** and **d**, one-way ANOVA was used to analyze significance. **f** Cells depleted for the indicated genes and expressing either GFP-CtIP or GFP-CtIP-SUMO1 were irradiated and RPA foci-positive cells was scored as in **b**. Values of RPA foci-positive cells were normalized to the percentage of S/G2 cells (Supplementary Fig. 6b), as in Fig. 1b. Statistically significant differences were analyzed using two-way ANOVA. For **b**–**e**, the average and SEM of at least three independent experiments were plotted. For all statistical analyses; *$P < 0.5$; **$P < 0.01$; ***$P < 0.001$

corresponding to a modified form of CtIP appeared among the purified sumoylated proteins, both with SUMO1 and SUMO2 isoforms (Supplementary Fig. 2). Interestingly, this CtIP modification was constitutive and occurred at similar levels both in the presence and the absence of DNA damage (Fig. 2d). Sumoylation of endogenous CtIP was confirmed in U2OS cell lines stably expressing 10 × His-SUMO1 (Fig. 2e). Moreover, CBX4 depletion reduced the amount of sumoylated endogenous CtIP (Fig. 2e), indicating that this E3 contributes to CtIP sumoylation.

**Sumoylation at Lys896 is essential for CtIP activity**. Analyzing in silico the amino acid sequence of CtIP using the GPS-SUMOsp 2.0 software, we found seven putative sumoylation sites (Fig. 3a, Supplementary Fig. 3a). We mutated all seven putative sumoylatable lysines to a non-modifiable arginine in the GFP-CtIP plasmid to produce the 7KR mutant. Importantly, this mutant was as defective in DNA end resection as the empty vector (Fig. 3b, Supplementary Fig. 3b, c). Only one of the predicted sumoylation sites (K896) was found in the carboxy-terminal part of the protein, the most critical portion of the protein in terms of DNA resection regulation that is weakly conserved between CtIP and its yeast homologs Sae2 and Ctp1. Moreover, such residue was very close to the Thr-847 essential phosphorylatable site[22] (Fig. 3a). Strikingly, arginine substitution of K896 (K896R) on its own caused complete impairment of DNA end resection (Fig. 3b). In contrast, only a mild effect was produced by the 6KR mutant,

in which all sites except K896 were mutated (Fig. 3b), strongly suggesting that the K896 residue is particularly relevant for DNA end processing. Similar results were obtained with the single-molecule analysis of resection tracks (SMART) assay, which measures the length of resected DNA in individual DNA fibers[26] (Fig. 3c), confirming that an intact lysine at position 896 of CtIP is crucial for proficient resection. This effect was not caused by cell cycle arrest, a general defect in protein folding or delocalization of CtIP from the nucleus (Supplementary Fig. 3d–f). Collectively, these data indicated that K896 of CtIP is indispensable for its function in resection.

Lysines are the substrate for many different PTMs besides sumoylation, such as ubiquitylation or acetylation[27]. Indeed, although mutation of all seven putative sumoylation sites in the 7KR mutant clearly reduced CtIP sumoylation with SUMO1 and SUMO2, the single K896R mutant was still sumoylated to a similar extent as wild-type CtIP (Supplementary Fig. 3g). This could reflect either that the loss of sumoylation at the K896R site was masked by sumoylation of other residues, or that the resection defect observed in the K896R mutant was independent of CtIP sumoylation. To confirm the relevance of sumoylation of K896, which is the previous to last amino acid at the C-terminus of the protein, we mimicked a constitutively sumoylated form of GFP-CtIP with a translational fusion of a non-conjugatable form of SUMO1 at the C-terminus of CtIP. We reasoned that the fusion protein CtIP K896R-SUMO1 (K896R-S) should compensate the phenotype shown by the mutant K896R if it was due to sumoylation defects. Indeed, we observed that the K896R-S

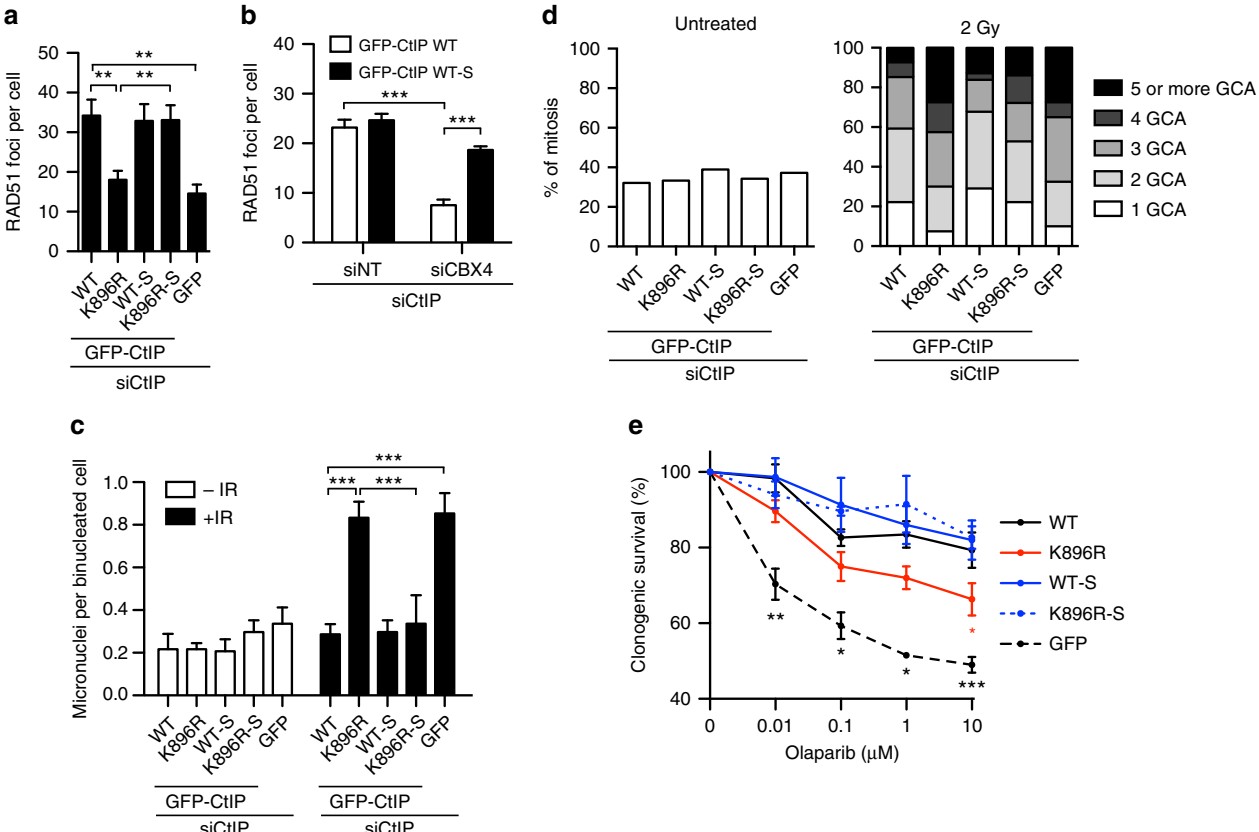

**Fig. 4** CtIP sumoylation at K896 contributes to genomic stability maintenance. **a** Average number of RAD51 foci per nuclei was analyzed 3 h after IR in U2OS cells stably expressing GFP or GFP-CtIP variants and knocked down for endogenous CtIP. RAD51 foci were scored in 100 cells per experiment. Average and SEM of three independent experiments is shown, and statistically significant differences were analyzed using one-way ANOVA. **b** Cells depleted for the indicated genes and expressing either GFP-CtIP or GFP-CtIP-SUMO1 were irradiated and RAD51 foci were scored as in **a**. Statistical significance was analyzed using two-way ANOVA. **c** Micronuclei appearance was scored in 100 binucleated U2OS cells per experiment upon IR or mock treatment. The average and SEM of three independent experiments is shown. One-way ANOVA analysis was performed within the irradiated samples. **d** Gross chromosomal aberrations (GCAs) were scored in GFP or GFP-CtIP expressing U2OS cells. The percentage of mitoses showing the different numbers of GCRs was calculated by analyzing 50 mitoses per experiment either in untreated (left) or treated cells with 2 Gy of IR. A representative experiment out of 3 independent repeats is shown. **e** Clonogenic survival of GFP or GFP-CtIP expressing U2OS cells upon acute treatment with the indicated dosed of the PARP inhibitor Olaparib. In each experiment, the average number of colonies per condition was scored in six technical replicates and normalized to the mock-treated condition. The average and SEM of three biological independent experiments is shown. Student's t-test between each condition and the wild-type form was performed. For all statistical analyses; *$P < 0.5$; **$P < 0.05$; ***$P < 0.001$

mutant was completely proficient for RPA foci formation (Fig. 3d, Supplementary Fig. 4). To unequivocally show that the effect of K896R mutation and SUMO1 fusion were indeed caused by affecting sumoylation at this sumoylation site, we mutated the key acidic amino acid E894 to alanine. Acidic residues are essential for recognition of the SUMO binding machinery of canonical and inverted sumoylation sites[28], so we reasoned that E894A mutant should phenocopy the K896R mutation, including the suppression by constitutive SUMO1 binding. Cells downregulated of endogenous CtIP but bearing a GFP-CtIP version mutated at Glu 894 were as deficient in DNA resection as the K896 mutant, but that defect disappeared when a SUMO1 sequence was transcriptionally fused to the construct (Fig. 3e, Supplementary Fig. 5). Strikingly, constitutive sumoylation of CtIP not only suppressed the K896R and E894 mutations in *cis*, but as well in *trans* the CBX4 depletion resection defect (Fig. 3f, Supplementary Fig. 6). Suppression of the CBX4 downregulation resection defect was highly specific, as it did not affect the minor resection defect observed after PIAS4 depletion (Fig. 3f). Thus, we conclude that CBX4's role in DNA end resection is mediated exclusively by the sumoylation of CtIP at K896, and that this modification is absolutely required for DNA end processing. As only a small

fraction of CtIP is normally sumoylated (Fig. 2d, e, Supplementary Fig. 2), we propose that this represents the only pool of CtIP protein available for DNA end resection.

**K896 sumoylation is required for genomic stability.** Defective sumoylation of CtIP by K896 or E894 mutations or CBX4 depletion seemed almost as defective in DNA end resection as CtIP depletion itself. We decided first to investigate the consequences for homologous recombination of impairing such PTM. We used the recruitment of the recombinase RAD51 by immunofluorescence as a proxy of homologous recombination. In agreement with a reduced DNA end resection, both K896R mutation and CBX4 depletion impaired recombination and reduced the formation of RAD51 foci (Fig. 4a, b, Supplementary Fig 7a, b). Strikingly, such defects were rescued by CtIP constitutive sumoylation (Fig. 4a, b, Supplementary Fig. 7a, b). Defective DNA repair usually leads to an increase in genomic instability. Indeed, a 4-fold increase in the irradiation-dependent accumulation of micronuclei could be observed in cells downregulated of endogenous CtIP and expressing GFP-CtIP-K896R mutant, a similar level of those cells expressing only GFP (Fig. 4c;

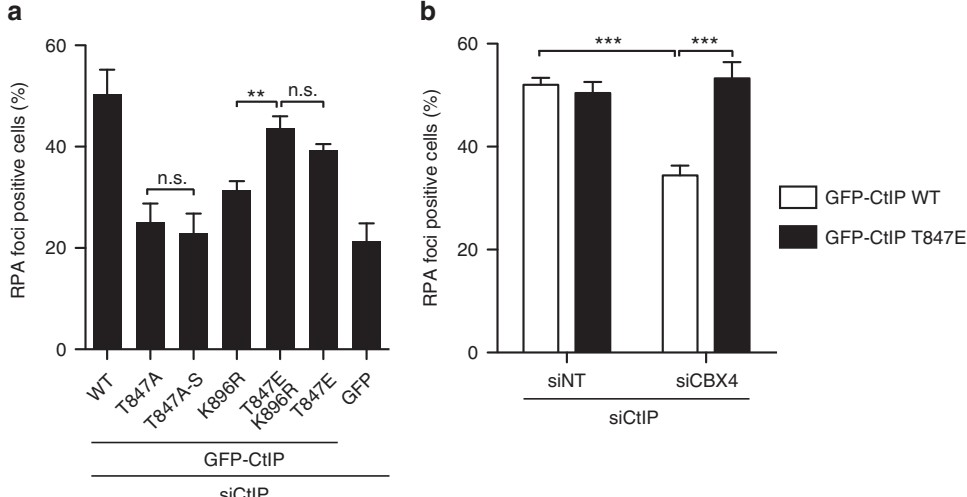

**Fig. 5** Phosphorylation of CtIP at T847 overcomes a lack of CBX4-mediated sumoylation for DNA resection. **a** RPA foci were measured 1 h after 10 Gy ionizing radiation in U2OS cells depleted of endogenous CtIP but expressing the indicated GFP-CtIP versions or GFP as a control. Statistically significance was determined with one-way ANOVA analysis, and only biological relevant comparisons are shown. **b** Same as **a** except that cells were co-transfected with a mix of siCtIP and siCBX4, or siNT as a control, as indicated. Values of RPA foci-positive cells were normalized to the percentage of S/G2 cells (Supplementary Fig. 8e), as in Figs. 1b and 3f. Statistically significance of differences was analyzed with two-way ANOVA. The average and SEM of three independent experiments were plotted in both panels. *$P < 0.5$; **$P < 0.05$; ***$P < 0.001$

see Supplementary Fig. 7c for an example of a micronucleus). As other phenotypes, this was completely suppressed when a SUMO1 sequence was fused to the construct (Fig. 4c). Interestingly, no increase in spontaneous micronuclei formation was observed in any cells. To complement this approach, we quantified gross chromosomal aberrations (GCAs) in cells bearing different CtIP variants using metaphase spreads. As expected, K896R mutation increases the number of GCA to a similar extent as the GFP-bearing control (Fig. 4d, Supplementary Fig. 7d). Such effect was almost completely reversed by constitutive CtIP sumoylation (Fig. 4d, Supplementary Fig. 7d).

Thus, GFP-CtIP-K896R mutation showed phenotypes similar to the cells expressing only GFP for DNA end resection, RAD51 accumulation and genomic instability. However, those defects caused only a mild cellular hyper-sensitivity to the PARP inhibitor (PARPi) Olaparib, a source of replication-dependent DSBs that greatly rely on homologous recombination for repair (Fig. 4e). Indeed, the differences were statistically significant only at high doses. Even less effect was observed in cells treated with camptothecin (Supplementary Fig. 7e). This suggested that CtIP might play additional roles on the repair of camptothecin and PARPi-generated DNA lesions that are independent of DNA end resection. As expected, the constitutively sumoylated CtIP behaved as well as or slightly better than the wildtype protein, even when K896 was mutated (Fig. 4e, Supplementary Fig. 7e).

**Crosstalk between CtIP sumoylation and CDK phosphorylation**. Among all known PTMs of CtIP, CDK-mediated phosphorylation at T847 has been reported to be essential for DNA end resection[22]. Due to the close proximity of T847 and K896, we wondered if they could be functionally related. We first ruled out that CDK phosphorylation was required for CtIP sumoylation (Supplementary Fig. 8a). We next analyzed RPA foci formation in cells expressing combinations of different SUMO- or phosphomutants (Fig. 5a, Supplementary Fig. 8b, c). As previously shown, the non-phosphorylatable T847A mutant was completely defective in terms of RPA foci formation, whereas the phosphomimetic mutant T847E was proficient[22] (Fig. 5a). Strikingly, constitutive phosphorylation at T847 (T847E mutant) suppressed

the phenotype of the K896R sumoylation mutant. However, constitutive sumoylation was not sufficient to overcome the defect in resection caused by the T847A phosphomutant (Fig. 5a, compare T847A with T847-S). These data suggest that the modifications occur sequentially, with CDK-mediated phosphorylation downstream of K896 sumoylation. Accordingly, the presence of the phosphomimetic version of CtIP completely abolished the CBX4-depletion phenotype of the resection defect but not the other phenotypes, such as the G1 arrest (Fig. 5b, Supplementary Fig. 8d, e).

**K896 sumoylation controls CtIP recruitment to damaged DNA**. We wondered how CtIP constitutive sumoylation might impact in its role on DNA repair. One tantalizing idea was that it controls its ability to be recruited to DSBs, explaining why only the sumoylated subpopulation of the protein can engage in DNA processing. Thus, we hypothesized that CtIP interaction with damaged DNA was modulated by sumoylation. To test this idea, we analyzed its recruitment to a doxycycline-inducible DSB created by the nuclease I-SceI[29]. This I-SceI cleavage site is flanked by 256 repeats of the lac operator DNA sequences (lacO), allowing the locus to be visualized at the lacO sequences by FISH (Fig. 6a). We compared the co-localization of either the DSB marker γH2AX or Flag-Cherry-CtIP with the lacO array in either a wild-type, a K896R, constitutive sumoylation or the combination of both mutations background (Fig. 6b–d, Supplementary Fig. 9). Upon induction of the break by doxycycline addition, we observed a twofold increase of wild-type CtIP (Fig. 6b, c). Strikingly, no increase in recruitment above background levels was observed in the K896R mutant (Fig. 6b, c). As before, CtIP recruitment was restored in a K896R mutant when a SUMO1 sequence was fused at the end of the protein (Fig. 6b, c). In fact, constitutive sumoylation of CtIP slightly increased its recruitment to the I-SceI induced break, even in combination with the K896R mutation, reinforcing the idea that sumoylation at the carboxy-terminal part of CtIP stimulates its recruitment to damaged chromatin (Fig. 6c). None of those differences were caused by a difference in DSB induction, as similar amounts of

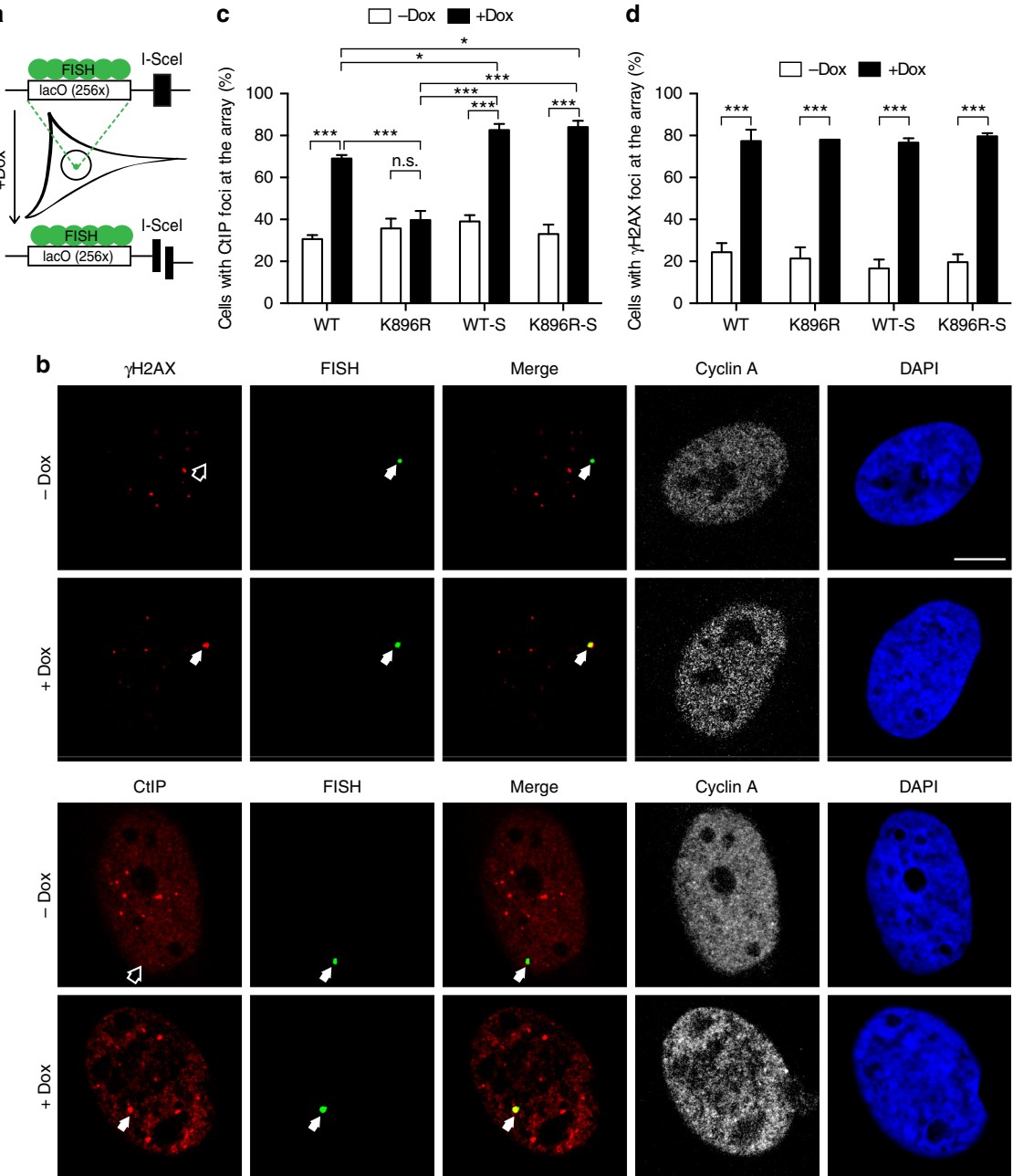

**Fig. 6** Sumoylation of CtIP controls its recruitment to DSBs. **a** Schematic representation of the experimental system to measure protein recruitment to DSBs. **b** Immuno-FISH representative confocal images of the lacO array in *green*, cyclin A in *white*, and γH2AX (*top*) or CtIP (*bottom*) in *red*, in U2OS19ptight13 expressing Flag-Cherry-CtIP. Cells were treated with doxycyline (+Dox) to create a DSB or mock treated (–Dox), as indicated in the Methods section. A *white arrow* marks the position of the array and the accumulation of either CtIP or γH2AX upon DSB induction. An *empty arrow* marks the position of the array when no accumulation of CtIP or γH2AX can be observed (undamaged conditions, –Dox). The *white scale bar* represents 7.5 µM. **c** Quantification of the percentage of colocalization of the lacO array with Flag-Cherry-CtIP variants in cells treated as mentioned in **b** depleted for endogenous CtIP. **d** Same as **c** but showing colocalization of the lacO array with γH2AX. Values in **c** and **d** represent the average and SEM of three independent experiments. En each experiment, 50 cells were scored. Two-way ANOVA analysis was performed to compare the indicated conditions in each graph. *$P < 0.5$; **$P < 0.01$; ***$P < 0.001$. When biologically relevant, not significant differences are marked as n.s

γH2AX recruitment were observed in cells expressing any of the CtIP variants, as measured by immuno-FISH (Fig. 6b, c).

## Discussion

Protein sumoylation is considered a critical way to fine-tune the cellular response to DNA damage, especially in the sensing and signaling of DSB[10]. This prompted us to try to resolve its implication in the modulation of the actual repair of the breaks, particularly at the level of DNA processing. Strong evidence in *Saccharomyces cerevisiae* suggested that sumoylation and resection are related. Indeed, many HR factors, including the essential resection factor MRX complex (Mre11–Rad50–Xrs2), are sumoylated in response to DNA damage and, strikingly, such sumoylation induction depends both on the MRX complex and DNA resection itself[20, 21]. In higher eukaryotes, however, the

relevance of SUMO conjugation in resection was less well established. Sumo ligases such as PIAS1 and PIAS4 are known to promote DSB repair and radioresistance, and, indeed, a two-fold reduction on RPA recruitment to laser lines in cells depleted of any of those E3s has been reported[18]. Interestingly, only PIAS4 depletion showed a similar effect in our hands, arguing that different types of lesions might have a distinct set of SUMO ligases requirements for proficient resection.

We decided to analyze a small subset of SUMO E3 specifically for their impact on resection. In order to compile the list of candidates, we reasoned that any that was described as nuclear or already related with the DDR might be involved. An important consideration when studying DNA resection is cell cycle. Extensive resection is only activated in S and G2 cells, most likely to avoid recombination when the sister chromatid is not present[4, 30]. In G1 cells, some breaks undergo limited DNA processing to allow alternative NHEJ to proceed[31]. However, this confined resection is slower and does not render a global RPA accumulation easily visible as RPA positive cells (see Supplementary Fig. 1 for examples of RPA positive and negative cells)[31]. Thus, cell cycle modifications rendered by the E3s depletion had to be accounted somehow. We normalized the RPA foci data with cell cycles profile. It is noteworthy that such approach did not result in any amplification of the difference with control cell, but the opposite. By such approach, we uncovered CBX4 as a critical factor for resection activation. CBX4 is a fine example of why considering cell cycle is so important in these studies. The effect of CBX4 was overestimated by the strong G1 accumulation in cells downregulated of this E3. However, even with the cell cycle correction, the effect was clear enough to make us suggest that CBX4 is required for DNA resection. We reason that the fact that constitutive sumoylated CtIP can rescue CBX4 depletion resection defect without suppressing the cell cycle arrest is proof enough that the lack of RPA positive cells is indeed due to hampered resection and not to an altered cell cycle profile.

Strikingly, the whole effect of CBX4 in DSB processing seems to rely on the sumoylation of a single substrate, CtIP, on a specific site, lysine 896. CtIP was known to be sumoylated in vitro[32] and our results agree with its sumoylation at multiple sites in vivo including, but not exclusively, at lysine 896. We still do not know the role of those other sumoylations, but the combined mutation of six of them mildly impaired resection. However, sumoylation impairment at K896 with either a K896R or a E894A mutation, completely abolishes CtIP activity on DNA processing. Sae2, CtIP functional counterpart in budding yeast, is also sumoylated at lysine 97 and this PTM is essential for proficient DNA resection in vivo[32]. However, in this case sumoylation occurs in response to DNA damage[32]. Commonly, sumoylation affect specific phenotypes, including the DDR ones, through the combined action of several substrates[11]. Thus, the individual contribution of specific sites is small. In stark contrast, CBX4 and CtIP resection roles rely exclusively on K896 sumoylation. This supports the idea that CtIP is the key regulator of DNA resection and that requires very specific PTMs to be activated[7]. This is similar to the CDK phosphorylation of CtIP at threonine 847, that acts as a binary ON/OFF switch[22] or the ATR or ATM-mediated phosphorylation of threonine 859[7]. Indeed, sumoylation and CDK- and ATR and/or ATM-mediated phosphorylation of CtIP parallelism does not finish here. Both sumoylation and phosphorylation of CtIP occur at multiple positions, and combination of several mutations of those sites inactivate the resection activity[7]. However, mutations on the closely spaced T847, T859, and K896 residues on their own completely abolish DNA resection[7, 22]. Considering the critical accumulation of specific PTMs at the carboxy-terminal part of the protein, it is not surprising that such region defines a key domain for resection regulation. Moreover, critical interactions with the resection factor MRN complex and the resection antagonist CCAR2 are also located in that region[8, 9]. Thus, the carboxy-terminal tail of CtIP seems to act as a hub in which several cellular signals converge in order to regulate DNA resection. Indeed, loss of such portion of the protein renders the protein inactive and acts as a dominant negative, causing the appearance of Seckel and Jawad syndromes[33]. We have shown that, in addition to the aforementioned interactions and PTMs, sumoylation at the very end of the protein is essential for its resection function. Strikingly, those PTMs, or at least T847 CDK phosphorylation and K896 sumoylation, are functionally connected and they seem to act sequentially rather that in parallel. Interestingly, budding yeast Sae2 is also sumoylated and phosphorylated by CDK, but in that case, they seem to act independently as double mutants Sae2-S267A K97A showed an additive effect in resection when compared with the single mutations[32]. Cascades of consecutive PTMs in proteins have been proposed before. For example, FEN1 nuclease sumoylation requires prior phosphorylation and in turn causes ubiquitylation[34]. Albeit we cannot observe such sequential appearance of sumoylation and phosphorylation, the fact that a mutation that mimics constitutive phosphorylation such as T847E can suppress both CtIP K896R mutant and CBX4 depletion, strongly agrees with this idea that SUMO conjugation predates CDK phosphorylation.

Only a small subpopulation of the protein is indeed conjugated to SUMO, as it is typical for sumoylation[13]. This low abundance of the modified form of the protein is usually attributed to local or temporal constrains. Most usually, sumoylation of proteins involved in dealing with DNA lesions are restricted to the proteins that are already recruited to damaged chromatin and/or induced by the presence of DNA damage[11]. In stark contrast, CtIP sumoylation on K896 seems to be a constitutive modification that predates the appearance of DNA damage and CtIP recruitment to DNA breaks. In that regard, it is very different to the DNA damage induced phosphorylation and more similar to the CDK-dependent modification of the protein. CtIP is known to have many different roles within the cell, including transcription regulation, replication, checkpoint activation or the G1/S transition[7]. We propose that sumoylation of CtIP defines a limited pool of the protein that can engage in resection. But, why limit the amount of CtIP that can be readily used to process DNA breaks? We reasoned that for most DNA breaks, NHEJ would suffice for repair, especially for clean breaks or the ones that appear in G1. Thus, limiting resection might be important to facilitate NHEJ on those circumstances. More importantly, resection might activate unscheduled single strand annealing or alt-NHEJ, both mutagenic repair pathways. So, by reducing the pool of CtIP that can be engaged in end processing, cells would modulate the decision between NHEJ and HR, but also the balance between different HR subpathways.

This resection control absolutely reflects in the appearance of recombination centers and in the formation of GCAs. But surprisingly, this does not correlate perfectly with cell survival to camptothecin or PARP inhibitor. This uncoupling between resection phenotype and cell survival to damaging agents have been observed before by us and others. For example, mutation of serine 327, that modulates CtIP interaction with BRCA1, showed mild resection defects but a hyper-sensitivity to DNA damaging agents similar to CtIP depletion[26, 35] and the T847E mutant of CtIP, that is resection proficient, is mildly sensitive to ionizing radiation[22]. Taken all together, these data seem to suggest that CtIP might play additional roles downstream in the DSB repair pathways that are independent on its resection activity and that K896R and S327A modification might act as separation of function mutants. However, we cannot exclude the possibility

that camptothecin and PARP inhibitor-induced DNA damage does not requires CtIP sumoylation due to the nature of the break. Along those lines, the nuclease activity is dispensable for clean breaks but essential for ragged ends[36].

Mechanistically, CtIP sumoylation controls resection at the level of CtIP recruitment to DNA damage. K896R mutant fails to be recruited to a single, I-SceI induced DSB, whereas the constitutively sumoylated form concentrates more avidly there. CtIP is known to bind DNA directly[36]. Interestingly, binding of several DDR factors to DNA is either increased or decreased by sumoylation[11]. Indeed, sumoylation of Tdp1, Top2, Yku70, and SLX4 stimulates their association with DNA[37–40]. However, we cannot conclude if the effect we observe is mediated by the direct binding to DNA or to damaged chromatin or even mediated by interaction with other factors. Sumoylation is a potent regulator of protein–protein interaction. In some cases, it might act as a molecular glue that stabilizes them[32]. Indeed, this has been documented for several DDR factors and, more relevant, for the MRX complex[11]. Interestingly, Mre11, the catalytic subunit of the MRX complex, possess several SUMO interacting motifs that are likely involved in MRX assembly[41]. Alternatively, sumoylation might disrupt the binding of CtIP to anti-resection factors, allowing CtIP recruitment or retention to DSBs. This would parallel what is observed in budding yeast, in which Sae2 is maintained in inactive aggregates and sumoylation of the protein solubilizes it[32].

In summary, CBX4 has emerged as a critical component of the DNA end resection machinery. Strikingly, and in contrast to other SUMO ligases previously described to act during the response to DNA damage, such as PIAS1 or PIAS4, CBX4 functions in a constitutive manner by sumoylating CtIP. We can circumscribe CBX4's role in resection to its sumoylation of a single factor at a single residue—namely, CtIP at K896. Our data suggest that further sumoylation occurs in CtIP, either by CBX4 or other E3 ligases, but that this has little or no effect on DNA end resection. We hypothesize that only the pool of CtIP that has been previously sumoylated by CBX4 at K896 can be recruited to DSBs, hence making it readily available for resection. We propose that the rest of the CtIP protein pool might be involved in its other functions, such as transcription and DNA replication[7]. By reducing the levels of CtIP that can be engaged in DNA end processing, the cells limit the availability of the resection machinery, thus regulating the balance between homologous recombination and NHEJ, a critical element in the maintenance of genomic integrity.

## Methods

**Cell lines and growth conditions**. U2OS (S. P. Jackson, University of Cambridge, UK), HEK293T (S. P. Jackson, University of Cambridge, UK), and U2OS 10 × His-SUMO1 cell lines (A. C. Vertegaal, Leiden University Medical Center) were grown in DMEM (Sigma-Aldrich) supplemented with 10% fetal bovine serum (Sigma-Aldrich), 2 mM L-glutamine (Sigma-Aldrich), 100 units ml$^{-1}$ penicillin, and 100 μg ml$^{-1}$ streptomycin (Sigma-Aldrich). U2OS stably expressing GFP-CtIP plasmids were grown in standard U2OS medium supplemented with 0.5 mg ml$^{-1}$ G418 (Gibco, Invitrogen). U2OS19ptight13 cells[42] were grown in the absence of phenol red tetracyclines, and supplemented with 0.8 mg ml$^{-1}$ G418. U2OS19ptight13 Flag-Cherry-CtIP populations were generated by transfection of the different plasmids and selection with 1 μg ml$^{-1}$ puromycin. Doxycyline (Sigma-Aldrich) was added to U2OS19ptight13 medium 14 h prior fixation to induce I-SceI cutting. Cells were tested for mycoplasma infection with MycoAlert PLUS Mycoplasma Detection kit (Lonza).

**siRNAs, plasmids and transfections**. siRNA duplexes were obtained from Sigma-Aldrich or Dharmacon (Supplementary Table 1) and were transfected using RNAiMax Lipofectamine Reagent Mix (Life Technologies), according to the manufacturer's instructions.

Plasmid transfection of U2OS cells was carried out using FuGENE 6 Transfection Reagent (Promega) according to the manufacturer's protocol.

HEK293T cells were transfected with a standard calcium phosphate transfection protocol.

pRSV 6 × His-SUMO1 and SUMO2 and GFP-CtIP plasmids were previously published[9, 43]. Point mutants of GFP-CtIP were produced using the QuickChange Lightning Site-Directed Mutagenesis Kit (Agilent Technologies), according to the manufacturer's instructions. To obtain GFP-CtIP-SUMO1 constructs, the stop codon of GFP-CtIP was substituted by a NotI restriction site, to have this sequence next to the BamHI site in which CtIP was cloned originally. SUMO1 cDNA was cloned from pRSV 6 × His-SUMO1 by PCR, adding NotI and BamHI sites to 5′ and 3′ ends of the sequence, respectively. The modified plasmid and the cloned sequence were then cut with NotI and BamHI and ligated. The SUMO1 sequence was introduced at the C-terminus of CtIP, thus leaving free the C-terminus of SUMO1. To make this SUMO unavailable for conjugation, the two terminal glycines were removed by mutagenesis. Flag-Cherry-CtIP plasmids (WT and K896R mutant) were assembled by golden gate cloning[44].

**Lentiviral infection**. A plasmid containing shRNA targeting CtIP (TRCN0000318738; Sigma) was transfected into HEK293T cells to produce lentiviruses as previously described[45]. Briefly, lentiviral particles were generated using 10 μg p8.91, 5 μg pVSV-G and 15 μg of the shRNA bearing plasmids by calcium phosphate transfection in A293T cells. After 48 h, lentiviruses were collected from the media by 100,000 × g centrifugation for 2 h at 4 °C. These lentiviruses were used to infect U2OS cells in the presence of 8 μg ml$^{-1}$ polybrene (Sigma), and CtIP knocked-down cells were then selected for 72 h with 1 μg ml$^{-1}$ puromycin.

**RT-qPCR**. RNA was extracted from U2OS cells using RNeasy Mini Kit (Qiagen), and cDNA was produced from RNA samples with QuantiTect Reverse Transcription Kit (Qiagen), according to the manufacturer's instructions. qPCR was performed using iTaq Universal SYBR Green Supermix (Bio-Rad) and the primers indicated in Supplementary Table 2.

**SDS-PAGE and western blot analysis**. Protein extracts were prepared in 2 × Laemmli buffer (4% SDS, 20% glycerol, 125 mM Tris-HCl, pH 6.8) and passed 10 times through a 0.5 mm needle-mounted syringe to reduce viscosity. Proteins were resolved by SDS-PAGE and transferred to low fluorescence PVDF membranes (Immobilon-FL, Millipore). Membranes were blocked with Odyssey Blocking Buffer (LI-COR) and blotted with the appropriate primary antibody and infrared dyed secondary antibodies (LI-COR) (Supplementary Table 3). Antibodies were prepared in blocking buffer supplemented with 0.1% Tween-20. Membranes were air-dried and scanned in an Odyssey Infrared Imaging System (LI-COR), and images were analyzed with ImageStudio software (LI-COR).

**His-tag pulldowns**. To perform His-tag pulldowns, HEK293T cells transiently transfected with pRSV 6 × His-SUMO or -SUMO2, or U2OS cells stably expressing 10 × His-SUMO1 or -SUMO2, were gently scraped off in PBS and lysed in binding buffer (20 mM Tris-HCl, pH 7.9, 150 mM NaCl, 8 M urea). Protein extracts (1 mg) were incubated with 50 μl of cobalt-based magnetic Dynabeads (Novex, Life Technologies) on a roller for 2 h at room temperature. Beads were then washed 3 × with binding buffer supplemented with 5 mM imidazol, and the final product was eluted by boiling the beads at 100 °C for 1 min with SDS sample buffer 2 × containing 200 mM imidazol.

**Immunoprecipitation**. For GFP immunoprecipitation, U2OS cells expressing GFP or GFP-CtIP were harvested in lysis buffer (50 mM Tris-HCl, pH 7.5, 50 mM NaCl, 1 mM EDTA, 0.2% Triton X-100, 1 × protease inhibitors [Roche] and 1 × phosphatase inhibitor cocktail 1 [Sigma]). Protein extract (1 mg) was mixed with 50 μl of washed magnetic anti-GFP beads (GFP-Trap_M, Chromotek) and incubated overnight at 4 °C with gentle rocking. Beads were then washed 3 × with lysis buffer, and the precipitate was eluted in SDS sample buffer by boiling the beads and loaded onto a gel.

To immunoprecipitate endogenous proteins, U2OS were scrapped in lysis buffer (50 mM Tris-HCl, pH 7.5, 50 mM NaCl, 1 mM EDTA, 0.5% NP-40, 1 × protease inhibitors [Roche] and 1 × phosphatase inhibitor cocktail 1 [Sigma]). To degrade DNA, 100 U ml$^{-1}$ Benzonase (Merck Millipore) was added to protein extracts and incubated for 30 min on ice. Protein extracts (1 mg) were then precleared with magnetic protein A Dynabeads (Novex) under gentle agitation at 4 °C for 30 min. Precleared samples were then incubated with anti-CBX4 antibody or a rabbit IgG (Sigma) as a control for 1.5 h at 4 °C and Dynabeads were added afterwards and were incubated for 1 h at 4 °C under gentle agitation. Beads were then washed three times with lysis buffer, and the precipitate was eluted in Laemmli buffer.

**Immunofluorescence, PLA, immuno-FISH and microscopy**. For RPA and RAD51 foci visualization, U2OS cells knocked-down for different proteins were seeded on coverslips. At 1 h (for RPA) or 3 h (for RAD51) after irradiation (10 Gy), coverslips were washed once with PBS followed by treatment with pre-extraction buffer (25 mM Tris-HCl, pH 7.5, 50 mM NaCl, 1 mM EDTA, 3 mM MgCl$_2$,

300 mM sucrose, and 0.2% Triton X-100) for 5 min on ice. Cells were fixed with 4% paraformaldehyde (w/v) in PBS for 20 min. Following three washes with PBS, cells were blocked for 1 h with 5% FBS in PBS, co-stained with the appropriate primary antibodies (Supplementary Table 3) in blocking solution overnight at 4 °C or for 2 h at room temperature, washed again with PBS and then co-immunostained with the appropriate secondary antibodies (Supplementary Table 3) in blocking buffer. After washing with PBS, coverslips were mounted into glass slides using Vectashield mounting medium with DAPI (Vector Laboratories).

For tubulin staining to count binucleated cells, fixation was performed with methanol for 10 min and acetone for 30 s on ice. Cells were then incubated with 5% FBS in PBS for 1 h, immunostained with anti-tubulin antibody for 2 h at room temperature, washed with PBS and incubated with the secondary antibody for 1 h at room temperature. Coverslips were mounted as described above.

For PLAs, U2OS cells were irradiated (10 Gy) or mock treated and fixed 1 h after with methanol for 10 min followed by acetone incubation for 30 s. After that, cells were blocked with blocking solution from the Duolink PLA Kit (Olink Bioscience) for 30 min at 37 °C and incubated with primary antibodies against CtIP and CBX4 (Supplementary Table 3) overnight at 4 °C. Samples were then incubated with MINUS and PLUS secondary PLA probes for 1 h at 37 °C, and the detection of these probes was carried out with the Duolink Detection Kit Red (Olink Bioscience). The number of PLA signals per nucleus was scored in at least 100 cells per sample.

RPA foci, RAD51 foci, and PLA immunofluorescences were analyzed using a Leica Fluorescence microscope.

For immuno-FISH, U2OS19ptight13 cells were fixed in 4% paraformaldehyde for 15 min, permeabilized in 0.5% Triton for 15 min, blocked in 3% BSA in PBS 0.1% Tween and incubated with primary and secondary antibodies (Supplementary Table 3) for 1 h each. Cells were then post fixed in 4% formaldehyde for 20 min, and the FISH protocol was carried out as previously described[29]. Briefly, samples were washed for 5 min in 2× SSC (saline-sodium citrate) buffer and 45 min in 2× SSC with a increasing temperature from room temperature to 72 °C, washed once in 70% ethanol and twice in absolute etanol. Coverslips were dried for 5 min at room temperatura and then incubated with 0.1 N NaOH for 10 min and washed in 2× SSC for 5 min. Coverslips were washed again in 70% ethanol and twice with absolute ethanol. After drying, cells were hybridized with a DNA probe for 30 s at 85 °C and incubated overnight at 37 °C. The probe was prepared by nick translation from the lacO-I-SceI plasmid. DNA probe (0.3 mg) was mixed with 9 mg of ssDNA and 3 mg of CotI human DNA (Roche) and precipitated with 2.53 vol of ethanol and 1/10 vol of 2.5 M sodium acetate for 30 min at −80 °C. After 20 min of centrifugation, the supernatant was discarded, and the pellet was washed with 70% ethanol and centrifuged again for 5 min. The supernatant was discarded, and the pellet was dried and resuspended in 20 ml of hybridization solution (50% formamide, 43 SSC, 10% dextran sulfate) per coverslip by vortexing for 1 h. The probe was denatured for 5 min at 90 °C and preannealed for at least 15 min at 37 °C before hybridization with cells.

The day after hybridization, immuno-FISH was revealed. Coverslips were washed twice for 20 min at 42 °C in 2× SSC and then incubated with secondary antibody and fluorescein anti-biotin (Vector Laboratories, SP-3040) at 1:100 dilution for 45 min. Vectashield Antifade mounting medium (Vector Laboratories, H-1200) containing DAPI. Immuno-FISH samples were analyzed using a confocal microscopy.

**Single-molecule analysis of resection tracks**. SMART was performed as described[26]. Briefly, cells were grown in the presence of 10 μM BrdU for 24 h. Cultures were then irradiated (10 Gy) and harvested after 1 h. Cells were embedded in low-melting agarose (Bio-Rad), followed by DNA extraction. DNA fibers were stretched on silanized coverslips, and immunofluorescence was carried out to detect BrdU (Supplementary Table 3). Samples were observed with a Nikon NI-E microscope, and images were taken and processed with the NIS ELEMENTS Nikon Software. For each experiment, at least 200 DNA fibers were analyzed, and the length of the fibers was measured with Adobe Photoshop CS4.

**Cell cycle analysis**. Cells were fixed with cold 70% ethanol overnight, incubated with 250 μg ml$^{-1}$ RNase A (Sigma) and 10 μg ml$^{-1}$ propidium iodide (Fluka) at 37 °C for 30 min and analyzed with a FACSCalibur (BD). Cell cycle distribution data were further analyzed using ModFit LT 3.0 software (Verity Software House Inc.).

**Proteinase K assay**. U2OS cells expressing GFP-tagged wild-type or K896R CtIP were subjected to GFP immunoprecipitation as described above. The immuno-precipitation product was divided into four parts, which were digested with increasing amounts of proteinase K (0, 5, 15 or 50 ng ml$^{-1}$) for exactly 3 min at room temperature. Samples were then boiled with SDS sample buffer for 5 min at 100 °C and loaded into a gradient SDS polyacrylamide gel.

**Cell fractionation**. U2OS cells expressing GFP-tagged wild-type or K896R CtIP were knocked down for endogenous CtIP by siRNA transfection and then, 48 h later, irradiated or not with 10 Gy ionizing radiation. At 1 h post irradiation, cells were harvested in resuspension buffer (10 mM HEPES, pH 7.9, 10 mM KCl,

1.5 mM MgCl$_2$, 0.34 M sucrose, 10% glycerol, 1 mM DTT and protease inhibitor cocktail). To disrupt cellular membranes, Triton X-100 was added to cell suspensions to a final concentration of 0.1%, and samples were incubated on ice for 8 min. Cells were then centrifuged at 4 °C at 1300 × g for 5 min to separate the cytoplasmic fraction from the nuclei pellet. Nuclei were then washed once with resuspension buffer and incubated with resuspension buffer supplemented with 90 U ml$^{-1}$ benzonase and 0.1 mg ml$^{-1}$ BSA for 1 h on ice. Cytoplasmic and nuclear fractions were boiled in SDS sample buffer and loaded into SDS polyacrylamide gel.

**Clonogenic survival assay**. U2OS expressing GFP and GFP-CtIP variants were transfected with siCtIP, seeded in low confluency and treated with Olaparib (AstraZeneca) at the indicated doses or mock treated with DMSO for 1 h. After treatment, cells were washed two times with PBS and cultured under standard conditions to allow colony formation for the next 8–10 days. Colonies were then counted after staining with 0.5% crystal violet/20% ethanol. The results were normalized to DMSO treatment.

**Micronuclei assay**. Cells were seeded onto coverslips and treated with 10 Gy ionizing radiation. Following treatment, cytochalasin B (Sigma) was added at 4 μg ml$^{-1}$. 24 h post treatment, cells were fixed and subjected to DAPI staining. Only binucleated cells were scored, which was confirmed by visualization of the cytoplasm with Tubulin immunofluorescence (performed as described in the immunofluorescence section).

**Chromosomal aberrations**. Aberrant chromosomes were counted on DAPI-stained mitotic spreads. Following CtIP depletion, GFP or GFP-CtIP expressing U2OS cells were exposed to 2 Gy of IR and then allowed to recover for 8 h in fresh medium before chromosome preparation. Within these 8 h, cells were treated with caffeine (2 mM) for the last 5 h to allow cells with gross chromosomal aberrations (GCAs) to enter mitosis, and for the last 3 h, they were treated with Colcemid (Sigma) to induce chromosome condensation. Cells were then harvested and treated with 0.075 M KCl for 10 min at 37 °C, fixed in methanol/acetic acid (3/1), washed twice with methanol/acetic acid (3/1), and then spread on a glass microscope slide, air-dried, and DAPI-stained.

**Statistical analysis**. Statistical significance was determined with a Student's t-test, one-way ANOVA or two-way ANOVA as indicated in each case, using PRISM software (Graphpad Software Inc.). Statistically significant differences were labeled with one, two or three asterisks if $P < 0.05$, $P < 0.01$ or $P < 0.001$, respectively.

**Data availability**. The data that support the findings of this study are available from the corresponding author.

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

## Acknowledgements

We thank A. Vertegaal and Mario García-Dominguez for providing reagents, Felipe Cortes-Ledesma and Felix Prado for critical reading of the manuscript, and Veronica Raker for style corrections. This work was funded by a R+D+I grant from the Spanish Ministry of Economy and Competitivity (SAF2010-14877) and an ERC Starting Grant (DSBRECA) to P.H. I.S.-B. is the recipient of a Ph.D. fellowship from the University of Sevilla (V Plan Propio) and received a EMBO short-term fellowship for this project.

## Author contributions

I.S.-B. performed all the experiments. C.C.-G. helped with the resection analysis of CtIP SUMO mutants and fusions and recruitment studies of CtIP. C.C.-R. did and analyzed the mitotic spreads. CtIP recruitment was performed by I.S.-B. in the E.S. laboratory with her help. B.R.-S.-M. and V.H. created the different CtIP constructs. P.H. supervised the work and wrote the manuscript. All authors read and commented on the manuscript.

## Additional information

**Competing interests:** The authors declare no competing financial interests.

