## [Peer Review file · Nature Communications]

Reviewers' comments:

Reviewer #1 (Remarks to the Author):

The authors report that sumoylation of CtIP by CBX4 is required for resection. They screened several sumoE3s and looked for reduced RPA foci as a sign of impaired resection. They then identify a lysine residue on CtIP which is sumoylated and when mutated impairs RPA foci formation. Lastly, they show that modification of CtIP is important for the recruitment of CtIP to DNA damage. From this the authors infer the importance of CBX4 dependent sumoylation of CtIP in resection during DSB repair.

CtIP is a key player in DSB repair by both homologous recombination but also in some forms of DNA end joining. The regulation of resection is a key step in preventing error prone repair and chromosome translocations that leads to cancer. The current manuscript, while addressing an interesting question is preliminary and does not fully establish its case.

Specific points:

The main feature of the study is the effect of sumo modification on resection, which relies heavily on the quantification of RPA foci. The authors do not indicate on their graphs or in the legends what the criteria are for a cell with RPA foci. How many foci does a cell need to have to be RPA positive? In the extended data it is clear that CtIP and CBX4 knockdown cells do have RPA foci, but not as many. The authors need to state what is considered an RPA positive cell.

The way the text reads suggests that cells were not subjected to DNA damage. Why then are the RPA foci levels very high? One might expect RPA foci formed in the absence of exogenous damage to be caused by ssDNA generated during replication, for which resection is not necessarily required. This seems strange and I feel like I am missing something.

Why were IP experiments not performed with antibodies against endogenous proteins. Overexpression of GFP tagged CtIP needs to be supported by data for endogenous protein.

The authors state that resection "is limited to the S and G2 phases of the cell cycle". This is not completely true as CtIP also functions in alt-NHEJ. With this in mind the normalization to S/G2 cells seems an artificial way of enhancing the observed effects.

Generally the reliance on RPA as a readout is fine to a point, but the manuscript would be improved by expanding the study to include broader biological effects of a resection defect. For example RAD51 recruitment, homologous recombination and on chromosomal aberrations/translocations should all be affected by a defect in sumoylation if the authors are correct.

Reviewer #2 (Remarks to the Author):

This story is intriguing and potentially important suggesting a single SUMOylation site of CtIP has a radical impact on resection. However it is incomplete. The manuscript presents no information on what the SUMOylation brings to CtIP or to resection.

CBX4 depletion results in reduced RPA foci. PLA suggest the interaction is not increased on IR, similarly CtIP is constitutively SUMOylated. One SUMOylation site K896 mutant caused impairment of resection (CtIP with the mutant is still SUMOylated). SUMO1 fused to the C-terminus) despite being the wrong way up) could overcome the need for CBX4 in a measure of resection.

Recruitment of CtIP to site of DSBs is partially dependent on the site (mutant CtIP at the array is already 30+ % even without Dox, only raising to 50% with dox, are these cells marked for cell cycle stage?).

Major concerns:

One is left without an understanding of the mechanisms involved here. It is not clear at all what SUMO brings to CtIP – particularly given that it is a constitutive modification.

Is the SUMO on CtIP central to an interaction that is required for recruitment or is it, as suggested, about interaction of CtIP with DNA. It is a very large jump between making the observation that the literature contains examples of SUMO increasing DNA interaction of factors and assuming that this is the mechanism here. There are many potential explanations.

If the modification is so crucial what is the impact of the CtIP mutants and CtIP-SUMO fusion on more critical measures of HR (RAD51 loading PARP-inhibitor sensitivity and chromosome integrity)?

The conclusions that the phosphorylation and SUMOylation are linked is premature. The data is also consistent with two independent events, in which the phosphorylation is functionally dominant. Since SUMOylation is constitutive it may come first – or, alternatively the populations maybe quite separate (given that phosphorylation is dominant) To argue co-dependence biochemistry is needed to show the SUMOylation status of phospho mutants and the phosphorylation status of SUMO site mutants.

As pointed out by the authors lysines are modified by several modifications. To strengthen the case a mutant of the glutamate of the PsiKxE of K896 should also be tested.

Minor concerns

Unfortunately there is no rescue shown for the CtIP- SUMO fusion and the images shown in fig 4b are not the critical ones of the experiment (the mutant comparison).

Is the statement on line 116 correct – “suppressed” appears contrary to the findings.

The RPA scoring for the E3 knockdowns is not explained. How was a cell scored as being RPA positive?

I'm concerned that RPA foci in S/G2 marked cells were not measured initially rather than trying to normalise RPA data from cells without markers on the basis of cell cycle skew (although I understand why this was done).

Reviewers' comments:

As an initial consideration for both reviewers, we realized that some of the points they addressed stemmed from the fact that the manuscript was written in a short format, as it was transferred from another journal that favour such style. However, now we have expanded the manuscript into a longer format with a proper introduction and a proper discussion that we think would clarify some of the referees' questions.

Reviewer #1 (Remarks to the Author):

The authors report that sumoylation of CtIP by CBX4 is required for resection. They screened several sumoE3s and looked for reduced RPA foci as a sign of impaired resection. They then identify a lysine residue on CtIP which is sumoylated and when mutated impairs RPA foci formation. Lastly, they show that modification of CtIP is important for the recruitment of CtIP to DNA damage. From this the authors infer the importance of CBX4 dependent sumoylation of CtIP in resection during DSB repair.

CtIP is a key player in DSB repair by both homologous recombination but also in some forms of DNA end joining. The regulation of resection is a key step in preventing error prone repair and chromosome translocations that leads to cancer. The current manuscript, while addressing an interesting question is preliminary and does not fully establish its case.

Specific points:

The main feature of the study is the effect of sumo modification on resection, which relies heavily on the quantification of RPA foci. The authors do not indicate on their graphs or in the legends what the criteria are for a cell with RPA foci. How many foci does a cell need to have to be RPA positive? In the extended data it is clear that CtIP and CBX4 knockdown cells do have RPA foci, but not as many. The authors need to state what is considered an RPA positive cell.

That is a very important point, and we apologize if it was not clear from the beginning. We have included several examples of what we consider a positive or a negative cell for RPA foci (see supplementary data 1c, red and white arrows, respectively). We have done so only in the control and CtIP depleted cells, for clarity of the figures but we can do so for every single RPA image in the text if asked. Moreover, we have now scored the number of RPA foci per cell upon CBX4 depletion automatically using MetaMorph software and those data have been included as well in the new Figure 1c.

The way the text reads suggests that cells were not subjected to DNA damage. Why then are the RPA foci levels very high? One might expect RPA foci formed in the absence of exogenous damage to be caused by ssDNA generated during replication, for which resection is not necessarily required. This seems strange and I feel like I am missing something.

Our apologies again for this serious oversight. The reviewer is absolutely right and the text was misleading. The cells were indeed irradiated on those experiments, but we did not mention it neither in the main text nor in the figure legends by mistake. We have

corrected the text and now it is stated in both places. Our apologies again and thank you very much for pointing that out.

Why were IP experiments not performed with antibodies against endogenous proteins. Overexpression of GFP tagged CtIP needs to be supported by data for endogenous protein.

New IP data using antibodies against endogenous proteins have been added as requested. The results agree with our previous observation with GFP-tagged CtIP and endogenous CtIP and CBX4 indeed interact (new figure 2b). Incidentally, the PLA data were also performed with endogenous proteins (currently in figure 2c).

The authors state that resection “is limited to the S and G2 phases of the cell cycle”. This is not completely true as CtIP also functions in alt-NHEJ. With this in mind the normalization to S/G2 cells seems an artificial way of enhancing the observed effects.

Again, the referee is right. We now state that extensive DNA resection is limited to S and G2. In the new discussion section, we discuss the role of limited resection for alt-NHEJ. Regarding the normalization, and the possibility of artificially enhancing the observed effects, it actually works the other way around. Although some resection can be observed in G1 (as seen by Markus Löbrich and colleagues), this is so limited that is fundamentally different from the S and G2 resection and most cells are RPA-foci negative unless you wait for longer times (see for example the recently published Biehs et al 2017). In fact, the normalization reduces the effect in all cases, including CBX4 depletion. It is worth noticing that the SUMO-mutants of CtIP do not change cell cycle profile and we still see the same and that constitutive sumoylation of CtIP rescues even CBX4 downregulation for the resection, but not cell cycle, phenotype. A more detailed explanation of the normalization and how it does not increase, but indeed reduces, the observed effect has been added to the text in the *results* and the *discussion* sections so the reader is aware of this.

Generally the reliance on RPA as a readout is fine to a point, but the manuscript would be improved by expanding the study to include broader biological effects of a resection defect. For example RAD51 recruitment, homologous recombination and on chromosomal aberrations/translocations should all be affected by a defect in sumoylation if the authors are correct.

First, we want to point out that we do not only rely on RPA foci for resection analysis. We also applied the SMART technique as well, that is an independent way to measure resection. In any case, the referee makes a fine point regarding other readouts to expand the study and the consequences of impairing CtIP sumoylation. Now we have included Rad51 foci, micronuclei accumulation, chromosomal aberrations and sensitivity to PARP inhibitor. We think the message is now more relevant and the paper has improved substantially thanks to this suggestion.

Reviewer #2 (Remarks to the Author):

This story is intriguing and potentially important suggesting a single SUMOylation site of CtIP has a radical impact on resection. However it is incomplete. The manuscript presents no information on what the SUMOylation brings to CtIP or to resection.

We understand that we do not have a complete story in terms of molecular mechanism, but our results present the compelling idea that not all CtIP is involved in resection, but only the SUMOylated form. Moreover, we can show, especially with the new data, that sumoylation controls CtIP recruitment. Strikingly, constitutive sumoylation causes a mild but significant increase in CtIP recruitment. Considering that the actual role of CtIP in resection, apart from the stimulation of MRN in a mysterious way, it is not known, it is really hard to pinpoint a more sophisticated role on that particular sumoylation. Indeed, nobody knows how T847 phosphorylation control CtIP function either. Both modifications at the carboxy terminal tail seem to be equally relevant. We hope that the new data and especially the expanded *discussion* will convince the referee that paper merits publication in Nat Communications, considering that in term of molecular mechanism we clearly show that sumoylation is essential for recruitment.

CBX4 depletion results in reduced RPA foci. PLA suggest the interaction is not increased on IR, similarly CtIP is constitutively SUMOylated. One SUMOylation site K896 mutant caused impairment of resection (CtIP with the mutant is still SUMOylated). SUMO1 fused to the C-terminus) despite being the wrong way up) could overcome the need for CBX4 in a measure of resection.

A fair summary. It is noteworthy that the original manuscript was transferred from another journal of the same editorial group and was in a report style, with little discussion. Now, we have taken full advantage of the Nat Comm format and expanded the discussion.

Recruitment of ctip to site of DSBs is partially dependent on the site (mutant CtIP at the array is already 30+ % even without Dox, only raising to 50% with dox, are these cells marked for cell cycle stage?).

We have done it both in cells labelled with Cyclin A and not labelled without Cyclin A and the results are the same. In the new figure 6 we included the Cyclin A stained cells for clarity. We do not know why the background level of CtIP binding is so high, maybe it has to do with the specific conformation of the LacO array, or the fact that the GFP-tagged CtIP constructs cause a mild overexpression of the protein. Even though, the DNA damage-induced recruitment is readily observed and it is completely dependent on CtIP sumoylation.

Major concerns:

One is left without an understanding of the mechanisms involved here. It is not clear at all what SUMO brings to CtIP – particularly given that it is a constitutive modification.

We have now expanded the discussion to clarify our model. Currently, we think it marks the subpopulation of the protein that can be engaged in resection by controlling its ability to be recruited to DSB. It is also clearly functionally related with CDK phosphorylation, but we do not know how. We hope now the mechanism we propose is stated more clearly. The fact that only sumoylated CtIP can be recruited to DNA damage and constitutive sumoylated CtIP binds even more avidly strongly support our model.

Is the SUMO on CtIP central to an interaction that is required for recruitment or is it, as suggested, about interaction of CtIP with DNA. It is a very large jump between making

the observation that the literature contains examples of SUMO increasing DNA interaction of factors and assuming that this is the mechanism here. There are many potential explanations.

As a rationale, we started on that point, but we agree that we do not have evidence of CtIP binding to DNA been altered or if this depends on other protein. We have toned down and discussed in more detail our findings. Again, as it is not completely clear how CtIP is recruited to DNA lesions (direct interaction with DNA, protein-protein interactions, a combination of both), it is hard to gain further mechanistic insights.

If the modification is so crucial what is the impact of the CtIP mutants and CtIP-SUMO fusion on more critical measures of HR (RAD51 loading PARP-inhibitor sensitivity and chromosome integrity)?

An excellent point. We have performed all three types of assay and all of them are affected by K896 sumoylation to a different extent. It is particularly striking the fact that Rad51 loading and chromosome integrity upon IR are completely dependent on that but PARP inhibitor sensitivity it is only mildly affected. This can be explained by different hypothesis that are now discussed in the new *discussion* section.

The conclusions that the phosphorylation and SUMOylation are linked is premature. The data is also consistent with two independent events, in which the phosphorylation is functionally dominant. Since SUMOylation is constitutive it may come first – or, alternatively the populations maybe quite separate (given that phosphorylation is dominant) To argue co-dependence biochemistry is needed to show the SUMOylation status of phospho mutants and the phosphorylation status of SUMO site mutants.

The referee is right. We could not see differences in sumoylation on the T847 mutants, but this might be masked by other sumoylation sites, a technical problem we have. Also, to demonstrate the effect of sumo mutants in the phosphorylation of T847 we will need phospho-specific antibodies. Unfortunately, the ones that exist are not so specific, and recognize CtIP CDK-phosphorylated at other CDK-sites (see the original Huertas and Jackson paper on JBC for details). However, genetically it is clear that once CtIP is phosphorylated it does not require sumoylation (as seen by the T847E K896R double mutant). We think this data are interesting enough to be included in the paper and support a sequential activation of CtIP that we will need to explore in following works. Is especially interesting when compared with budding yeast SAE2, that is also sumoylated and phosphorylated by CDK, but in this case these modifications act independently (see new *discussion*)

As pointed out by the authors lysines are modified by several modifications. To strengthen the case a mutant of the glutamate of the PsiKxE of K896 should also be tested.

A fair point. In this case, the SUMO site is a non-canonical one in which the acidic residue, in this case Glutamate, is upstream in the sequence (E894) (Matic et al 2010). The analysis of that mutant is now included. Not only it does impair resection, but it is suppressed by the addition of a SUMO1 at the end of the protein (upside down as mentioned by the referee). We want to thank the reviewer for this comment, as these

experiments have strengthened the idea that the resection phenotype is due to sumoylation impairment.

Minor concerns

Unfortunately there is no rescue shown for the CtIP- SUMO fusion and the images shown in fig 4b are not the critical ones of the experiment (the mutant comparison).

The rescue has now been included. We only included a WT as an example of how the recruitment does look for space and clarity sake. Including five images: red (the recruited protein), green (the LacO array), red and green merge, grey (cyclin A) and DAPI, for all four Cherry-FLAG constructs, in two conditions (+/- DOX), for CtIP and gammaH2AX, i.e. 5x4x2x2=80 images, create a huge and impractical figure. Bear in mind that in all cases we can observe cells positive or negative for the recruitment, and the focus itself is identical in all Cherry-FLAG-CtIP variants. Thus, the important bit is the quantification included in panel 6c and 6d. In our opinion, such figure will make sense if we claimed that the recruited protein looked different somehow. However, we are including a figure for the reviewer consideration with all the images and if the reviewer and/or the editor think it is absolutely essential to include in the paper, either as a main or supplementary figure, we will be happy to do so. For the figure, see below at the end of this letter.

Is the statement on line 116 correct – “suppressed” appears contrary to the findings.

Thanks for pointing out that the sentence can be misleading. Our intention was to state that constitutive sumoylation suppressed K896R phenotype. We have changed the sentence to “we observed that the K896R-S mutant was completely proficient for RPA foci formation”.

The RPA scoring for the E3 knockdowns is not explained. How was a cell scored as being RPA positive?

Examples of RPA positive and RPA negative cells have been labelled with red and white arrows in supplementary data 1c. Moreover, a computer-based scoring of the number of RPA foci per cell for CBX4 depletion has been included to avoid any suspicion of a bias in the quantification.

I’m concerned that RPA foci in S/G2 marked cells were not measured initially rather than trying to normalise RPA data from cells without markers on the basis of cell cycle skew (although I understand why this was done).

There are two problems with cell cycle labelling for the RPA-foci experiments with E3 ligases. First, to observe RPA foci by IF pre-extraction is required. This makes all the markers not tightly bound to chromatin (such as Cyclin A) to be washed away in the process. Other markers as CENPF might be used, but it is always possible that any of those markers is in turn affected by some of those E3 ligases. It is a bit of a technical conundrum that we tried to solve to the best of our possibilities. It is noteworthy that the type of normalization we do not only does not increase the effect of depletion of any of those E3 in resection, but it actually reduces it. Moreover, CtIP-SUMO1 fusion rescues CBX4 phenotypes on resection without changing cell cycle profiles, supporting the

effect we see is not due to the normalization itself. Altogether, and with the obvious technical concerns about missing some mild effect of other E3s, we think considering all the data it is clear that CBX4 affects resection through sumoylation of CtIP at K896. For us, this message is supported by the data we have and it is relevant enough to grant publication in Nat Comm. An explanation of the effect of normalization in RPA foci has been included in the *results* section, and a more in-depth discussion of what does represent it is now present in the new *discussion* section, so the reader is aware of it all the time.

REVIEWERS' COMMENTS:

Reviewer #1 (Remarks to the Author):

The manuscript is much improved in terms of its experimental content and completeness. The discussion section is now overlong as the authors have attempted to justify their approach and data. This might be shortened to discuss only relevant issues relating the current work to the literature.

It is noted that the criteria for % cells with RPA foci was not defined. The authors improved their quantification by using software to measure RPA foci per cell. But Figs 1 A and B score cells as RPA positive or not. The images provided show clear differences between RPA foci induced cells and those not. Nevertheless for quantification purposes the authors must define how many foci they count to score the cell as positive >1, >5, >10 etc.

The main weakness of the manuscript is still a lack of mechanism. The authors discuss a sub-population of resection permissible CtIP but they do not attempt to test this. The outcome is that they authors have an interesting observation that sumo controls resection but little understanding of how, especially as this regulation is constitutive and resection is damage inducible.

Reviewer #2 (Remarks to the Author):

In this revision the authors have made considerable improvements to the manuscript. While the central question of what it is SUMO brings to the molecular biology of CtIP remains unanswered the manuscript is now in a journal suitable for the level of answers the group have arrived at. I would agree with them that the efforts to squeeze the previous version of the manuscript into a short report format did them no favors - where as here it is well presented.

The expansion of the discussion is particularly welcome as it addresses the questions that remain open, provides some intriguing speculation about their own data and the literature and also is able to take the time to point out the difficulties of some of the work undertaken (the initial screen for example). The rebuttal letter is also very well written.

A mystery that arises from the new data is the disconnect observed between the resection failure seen and a only slight impact on PARP- inhibitor sensitivity -which is hard to understand in the context of poor resection. Indeed I would have expected a greater sensitivity with CtIP loss and GFP only (the scale is not a log one). (Note: The figure legend 4e does not state that the endogenous CtIP is depleted -although I assume it was from the data).

There are several explanations for the odd finding. Could the drug be partially inactive or the knock-down slight? My one significant suggestion is that this experiment be repeated with new olaparib and -separately- with Camptothecin to check this surprising result.

Nevertheless notably the defect that there is with K896R-CtIP is rescued with K896R-SUMO.

Overall the data is well presented and justify the claims of the paper. The paper is well written and convincing and the findings novel and interesting.

First, we would like to thank both reviewers for taking the time to review for a second time our manuscript. In relationship to the new issues they have addressed, a point-by-point response follows:

Reviewer #1 (Remarks to the Author):

The manuscript is much improved in terms of its experimental content and completeness. The discussion section is now overlong as the authors have attempted to justify their approach and data. This might be shortened to discuss only relevant issues relating the current work to the literature.

Thanks a lot for the positive feedback. As reviewer 2 had a different view about the length of the discussion (he or she states that "The expansion of the discussion is particularly welcome as it addresses the questions that remain open"), we have contacted the editor to solve this issue. Considering both reviewers opinion, together with the correspondence with the editor, we have decided to let the discussion as it is.

It is noted that the criteria for % cells with RPA foci was not defined. The authors improved their quantification by using software to measure RPA foci per cell. But Figs 1 A and B score cells as RPA positive or not. The images provided show clear differences between RPA foci induced cells and those not. Nevertheless for quantification purposes the authors must define how many foci they count to score the cell as positive >1, >5, >10 etc.

This is now included in the main text the first time we use RPA foci as a readout of resection (Figure 1a).

The main weakness of the manuscript is still a lack of mechanism. The authors discuss a sub-population of resection permissible CtIP but they do not attempt to test this. The outcome is that they authors have an interesting observation that sumo controls resection but little understanding of how, especially as this regulation is constitutive and resection is damage inducible.

We understand the referee reservation, but still considered we have enough mechanistical insight, i.e. CtIP sumoylation controls its recruitment to DSBs, to grant publication in Nat communication.

Reviewer #2 (Remarks to the Author):

In this revision the authors have made considerable improvements to the manuscript. While the central question of what it is SUMO brings to the molecular biology of CtIP remains unanswered the manuscript is now in a journal suitable for the level of answers the group have arrived at. I would agree with them that the efforts to squeeze the previous version of the manuscript into a short report format did them no favors - where as here it is well presented.

The expansion of the discussion is particularly welcome as it addresses the questions that remain open, provides some intriguing speculation about their own data and the literature and also is able to take the time to point out the difficulties of some of the work undertaken (the initial screen for example). The rebuttal letter is also very well written.

A mystery that arises from the new data is the disconnect observed between the resection failure seen and a only slight impact on PARP- inhibitor sensitivity -which is hard to understand in the context of poor resection. Indeed I would have expected a greater sensitivity with CtIP loss and GFP only (the scale is not a log one). (Note: The figure legend 4e does not state that the endogenous CtIP is depleted -although I assume it was from the data).

First, we apologize for not stating in the figure legend that the cells were depleted for endogenous CtIP. Now it is included. Second, this small effect on the GFP only is a technical issue with the fact that we are using a complementation system. U2OS depleted for CtIP expressing GFP-CtIP does not behave hundred percent as normal U2OS. That means that our wildtype is mildly sensitive to DSB inducing agents, if compared with U2OS, so the differences are less obvious with the GFP only control. Even do, that is what we normally find in this complementation systems.

There are several explanations for the odd finding. Could the drug be partially inactive or the knock-down slight? My one significant suggestion is that this experiment be repeated with new olaparib and - separately- with Camptothecin to check this surprising result.

We have repeated the experiments with ne Olaparib with similar results. Moreover, the results with camptothecin agree with no effect of K896 sumoylation in cell survival in U2OS. These data are now included in the new version of the manuscript (new supplementary Fig. 7e).

Nevertheless notably the defect that there is with K896R-CtIP is rescued with K896R-SUMO.

Overall the data is well presented and justify the claims of the paper. The paper is well written and convincing and the findings novel and interesting.

Thank you very much for your kind revision.